# The cost-of-illness trend of schizophrenia in South Korea from 2006 to 2016

Minkyung Jo[1,2], Hyun-Jin Kim[3], Soo Jung Rim[1,4], Min Geu Lee[1‡], Chul Eung Kim[1‡], Subin Park[1]*

1 Department of Research Planning, Mental Health Research Institute, National Center for Mental Health, Seoul, South Korea, 2 Graduate School of Public Health, Korea University, Seoul, South Korea, 3 Department of Clinical Research for Rehabilitation, National Rehabilitation Research Institute, National Rehabilitation Center, Seoul, South Korea, 4 Graduate School of Psychology, Seoul National University, Seoul, South Korea

☉ These authors contributed equally to this work.
‡ These authors also contributed equally to this work.
* subin21@korea.kr

**Data Availability Statement:** There are ethical and legal restrictions (i.e., data contains potentially sensitive information and personally identifiable information) since the data contains sensitive

## Abstract

Globally, about one in four people develop a psychiatric disorder during their lifetime. Specifically, the lifetime prevalence of schizophrenia is about 0.48%, and schizophrenia can have detrimental effects on a patient's life. Therefore, estimating the economic burden of schizophrenia is important. We investigated the cost-of-illness trend of schizophrenia in South Korea from 2006 to 2016. The cost-of-illness trend was estimated from a societal perspective using a prevalence-based approach for direct costs and a human capital approach for indirect costs. We utilized information from the following sources: 1) National Health Insurance Service, 2) Korean Statistical Information Service, Statistics Korea, 3) the National Survey of Persons with Disabilities, 4) Budget and Fund Operation Plan, Ministry of Justice, 5) Budget and Fund Operation Plan, Ministry of Health and Welfare, and 6) annual reports from the National Mental Health Welfare Commission. Direct healthcare costs, direct non-healthcare costs, and indirect costs by sex and age group were calculated along with sensitivity analyses of the estimates. The cost-of-illness of schizophrenia in Korea steadily increased from 2006 to 2016, with most costs being indirect costs. Individuals in their 40s and 50s accounted for most of the direct and indirect costs. Among indirect costs, the costs due to unemployment were most prevalent. Our estimation implies that schizophrenia is associated with a vast cost-of-illness in Korea. Policymakers, researchers, and physicians need to put effort into shortening the duration of untreated psychosis, guide patients to receive community-care-based services rather than hospital-based services and empower lay people to learn about schizophrenia.

## Introduction

Many individuals have mental disorders globally, and one in four people will develop a mental disorder during their lifetime [1]. Specifically, the lifetime and 12-month prevalence rate of

patient information. However, the data is available from NHIS (National Health Insurance System) upon request to the 'Information analysis department in Big data operation room'. Researchers who may wish to get access to the data could contact the Bid data operation room (+82-33-736-2432). Also, the url for the website is: https://nhiss.nhis.or.kr/bd/ab/bdaba000eng.do; jsessionid=cK18Qv01aQU6z3lIVw344vo

having any psychiatric disorder in South Korea (Korea from hereon) is 25.4% and 11.9%, respectively [2]. Moreover, the global burden of mental disorders comprises about 30% of years lived with disability and 15% of disability-adjusted life years [3]. These numbers indicate that improving mental health is a major aspect of increasing one's wellbeing.

Schizophrenia is a psychiatric disorder that includes symptoms such as delusions, hallucinations, and disorganized speech and behavior [4]. According to a systematic literature review [5], the median 12-month and lifetime prevalence rate of schizophrenia is 0.33% (interquartile range = 0.26%–0.51%) and 0.48% (interquartile range = 0.34–0.85%), respectively. Despite the low prevalence rate, schizophrenia has detrimental effects on patients' lives, and it places tremendous health, social, and economic burdens on patients, families, caregivers, and society [6]. For instance, patients with schizophrenia have low quality of life [7], impeded employment [8], can be involved in violence [9], have a high risk of acquiring physical diseases (e.g., cardiovascular disease) [10], and have a reduced life expectancy [11]. Consequently, it is no surprise that schizophrenia is in the top 15 (of 328) of diseases and injuries included in the "leading cause of years lived with disability" [12].

Notably, the Korean government has increased the national health expenditures allocated to mental health [13]. Despite the increase, only 2.6% of the total expenditures was allocated to mental health in 2014 [14]. Moreover, the amount is only about 2% of what the U.S. allocates to mental health [15]. Accordingly, it is important to determine what area requires more funding.

Korea provides universal healthcare through National Health Insurance (NHI) and Medical Aid programs. The NHI covers about 97% of the population and the rest is covered by Medical Aid [16]. Consequently, it is possible to construct a health insurance database that contains key data such as health records, prescriptions, etc. Moreover, the NHI Service (NHIS) provide these data to researchers for research purposes.

Investigating the cost-of-illness (COI) of schizophrenia is needed to establish proper and effective healthcare policy and intervention [17]. However, scant studies have investigated the economic burden of schizophrenia in Korea. A previous study [18] regarding the economic burden of schizophrenia (estimated from a societal perspective) in Korea is dated, and other studies [19,20] utilized disability-adjusted life years when estimating the economic cost. Moreover, no study has investigated the economic burden trend regarding schizophrenia in Korea. Therefore, from a societal perspective, we examined the COI trend of schizophrenia in Korea from 2006 to 2016.

## Materials and methods

### Study design

COI studies estimate the direct and indirect costs intertwined with a specific disease [16]. Direct costs are disease-related payments, which are divided into two parts: direct healthcare costs and direct non-healthcare costs. Indirect costs account for resources that are lost owing to the disease. Therefore, we calculated the cost for three main categories: direct healthcare costs, direct non-healthcare costs, and indirect costs related to those being treated for schizophrenia in Korea. When converting Korean Won to U.S. dollars, we used the average exchange rate of each year. The prevalence approach was utilized for direct costs, and the human capital approach was used for indirect costs.

The requirement for written, informed consent was waived. This study was approved by the institutional review board of the National Center for Mental Health, Korea (no. 116271-2017-10).

We utilized the following sources: 1) claims data and a medical expense survey from the NHIS [21]; 2) Korean Statistical Information Service, Statistics Korea [22]; 3) the National Survey of Persons with Disabilities [23,24]; 4) Budget and Fund Operation Plan, Ministry of Justice [25]; 5) Budget and Fund Operation Plan, Ministry of Health and Welfare [26]; and 6) annual reports of the National Mental Health Welfare Commission [27].

## Direct healthcare costs

Direct healthcare costs are defined as medical care costs, which include inpatient care costs, outpatient treatment costs, pharmacy costs, and non-covered care costs. The information regarding inpatient treatment costs, outpatient treatment costs, and pharmacy costs were obtained from the NHIS claims data from 2006 to 2016 [21]. However, the non-covered medical expenses were not retrievable from NHIS; therefore, these were estimated by the proportion of the non-covered care costs (i.e., 4.3% for inpatient and 2.3% for outpatient care) regarding mental health from the 2009 medical expense survey [28].

## Direct non-healthcare costs

Direct non-healthcare costs are defined as costs that are associated with patient management, other than direct healthcare costs. We included the following direct non-healthcare costs in this study: incarceration costs, community mental healthcare center costs, sanatoria costs, rehabilitation facility costs, and transport costs [2]. Patients with schizophrenia are involved in crime (mediated by substance use comorbidity) (mediated by substance use comorbidity) more so than is the general population [9]. Accordingly, there are various costs that are related to this issue such as policing and prosecution costs. However, those specific data were not available. Instead, we focused on costs that are retrievable—incarceration cost. Incarceration cost was calculated based on the Budget and Fund Operation Plan, Ministry of Justice [25]. Patients do not receive a diagnosis or prescription at a community mental health center; rather, patients visit community mental healthcare centers for rehabilitation purposes. In addition, most community mental healthcare costs are covered by the government, not by the individual. Therefore, community mental healthcare center costs were considered as direct non-healthcare costs.

Community mental healthcare center costs were estimated using information included in the budget of the local mental health centers [27]. The cost for sanatoria was based on the Budget and Fund Operation Plan, Ministry of Health and Welfare [26]. Psychiatric rehabilitation is vital for treatment as it assists patients with reaching their optimal level of functioning [29]. Therefore, the costs for rehabilitation facilities were also estimated through the Budget and Fund Operation Plan, Ministry of Health and Welfare [26]. The community mental healthcare center costs, sanatoria costs, and psychiatric rehabilitation costs were estimated by the annual operating costs from each report (e.g., annual operating cost of community mental healthcare center by local mental health centers in 2007 * treated prevalence rate of schizophrenia in 2007). We estimated the transport cost by multiplying the total number of hospital visits by the mean return fare [18].

## Indirect costs

Indirect costs included productivity loss due to unemployment, workplace productivity loss due to morbidity, premature death costs, and caregiving costs. Information regarding indirect costs was retrieved from Statistics Korea [22], the NHIS [21], and the National Survey of Persons with Disabilities [23,24]. There has been a shift with caregiving responsibility of those with mental disorders—from hospital-based to family- or caregiver-based [30]. Therefore, it is

important to estimate the caregiving costs associated with schizophrenia. The information regarding caregiving costs was estimated by information provided by Statistics Korea [22], the National Survey of Persons with Disabilities [23,24], and the NHIS [21]. For the exact equations for indirect costs, see S1 Table.

### Sensitivity analyses

Some costs were not attributable directly (e.g., non-covered medical expenses). Therefore, we conducted a sensitivity analysis to evaluate the robustness of the estimated costs utilizing the lowest (11%) and highest (20%) proportion of non-covered medical expenses reported by NHIS [31]. In addition, the discount rates (0% and 5%) of expected future earnings and the productivity costs due to premature mortality (using the friction-cost method for 1 month, 3 months, and 6 months) were calculated. A more-detailed description of the method regarding sensitivity analyses can be found elsewhere [18]. All statistical analyses were performed with SAS 9.4 (SAS Institute Inc., Cary, North Carolina, USA).

## Results

### Treated prevalence of schizophrenia in Korea from 2006 to 2016

The numbers of patients who were treated for schizophrenia, those registered as having a mental disability, and those who died from schizophrenia (i.e., cause of death was schizophrenia) or committed suicide each year from 2006 to 2016 are presented in Table 1. The treated prevalence rate has increased from 0.33% to 0.40% during this period. In total, 448,490 patients (when duplicates were removed) were treated for schizophrenia from 2006 to 2016 in Korea. Additionally, 109,306 individuals (when duplicates were removed) were classified as having mental disability among those treated for schizophrenia during that period. Moreover, about 1,000 people each year died from schizophrenia or committed suicide.

### COI of schizophrenia each year by each category

The total COI of schizophrenia and specific costs for each category—direct healthcare costs, direct non-healthcare costs, and indirect costs—are presented in Table 2. The total direct healthcare cost was about 451 billion Won in 2006 and rose to approximately 868 billion Won

**Table 1. Individuals treated for schizophrenia, registered as having a mental disability, or who died from schizophrenia/committed suicide from 2006 to 2016 in Korea.**

|  | Treated for schizophrenia | | Determined to have a mental disability | | Died from schizophrenia or suicide | |
|---|---|---|---|---|---|---|
|  | n | % | n | % | n | % |
| 2006 | 160,157 | 0.33 | 32,300 | 0.20 | 631 | 0.00 |
| 2007 | 165,544 | 0.34 | 48,536 | 0.29 | 926 | 0.01 |
| 2008 | 171,663 | 0.35 | 58,168 | 0.34 | 1,008 | 0.01 |
| 2009 | 175,230 | 0.35 | 64,824 | 0.37 | 1,054 | 0.01 |
| 2010 | 174,492 | 0.35 | 69,468 | 0.40 | 1,066 | 0.01 |
| 2011 | 179,254 | 0.36 | 73,664 | 0.41 | 1,122 | 0.01 |
| 2012 | 187,390 | 0.37 | 76,548 | 0.41 | 1,091 | 0.01 |
| 2013 | 189,961 | 0.38 | 78,260 | 0.41 | 1,093 | 0.01 |
| 2014 | 191,502 | 0.38 | 79,694 | 0.42 | 1,063 | 0.01 |
| 2015 | 199,743 | 0.39 | 82,058 | 0.41 | 1,138 | 0.01 |
| 2016 | 202,345 | 0.40 | 82,976 | 0.41 | 1,110 | 0.01 |

**Table 2. Overall cost of schizophrenia in Korea from 2006 to 2016.**

| Cost type | | | 2006 | % | 2007 | % | 2008 | % | 2009 | % | 2010 | % | 2011 | % | 2012 | % | 2013 | % | 2014 | % | 2015 | % | 2016 | % |
|---|---|---|---|---|---|---|---|---|---|---|---|---|---|---|---|---|---|---|---|---|---|---|---|---|
| Direct cost | Direct healthcare | Inpatient care costs | 325,324 | 75% | 361,527 | 75% | 399,306 | 76% | 486,348 | 78% | 528,291 | 78% | 567,542 | 79% | 598,032 | 80% | 619,228 | 80% | 633,385 | 81% | 660,052 | 81% | 674,963 | 81% |
| | | uncovered inpatient care cost | 13,289 | | 14,646 | | 15,926 | | 20,033 | | 21,977 | | 23,617 | | 24,829 | | 25,743 | | 26,334 | | 27,375 | | 27,950 | |
| | | Outpatient care costs | 93,095 | 21% | 101,722 | 21% | 109,056 | 20% | 118,553 | 19% | 125,385 | 18% | 128,384 | 18% | 126,982 | 17% | 132,692 | 17% | 134,319 | 17% | 136,327 | 16% | 140,672 | 17% |
| | | uncovered outpatient care cost | 1,825 | | 1,994 | | 2,148 | | 2,425 | | 2,605 | | 2,674 | | 2,629 | | 2,741 | | 2,779 | | 2,805 | | 2,887 | |
| | | Pharmacy costs | 17,583 | 4% | 19,807 | 4% | 21,677 | 4% | 23,075 | 4% | 23,831 | 3% | 23,895 | 3% | 23,845 | 3% | 23,535 | 3% | 21,955 | 3% | 22,078 | 3% | 22,221 | 3% |
| | | Total | 451,116 | 12% | 499,695 | 12% | 548,114 | 12% | 650,434 | 13% | 702,089 | 14% | 746,111 | 14% | 776,316 | 13% | 803,939 | 13% | 818,772 | 13% | 848,637 | 13% | 868,693 | 13% |
| | Direct non-healthcare | Incarceration | 1,223 | 1% | 1,261 | 1% | 1,540 | 1% | 1,545 | 1% | 1,901 | 1% | 1,813 | 1% | 2,023 | 1% | 2,530 | 1% | 2,673 | 1% | 3,471 | 1% | 3,969 | 2% |
| | | Community mental healthcare centers | 4,149 | 3% | 3,966 | 3% | 4,347 | 3% | 4,738 | 3% | 4,657 | 3% | 5,033 | 3% | 5,357 | 3% | 6,065 | 3% | 6,537 | 3% | 17,113 | 7% | 16,902 | 7% |
| | | Sanatoria | 81,443 | 60% | 84,847 | 60% | 86,254 | 59% | 89,653 | 59% | 92,319 | 59% | 95,507 | 59% | 129,532 | 63% | 115,482 | 57% | 118,509 | 56% | 122,238 | 52% | 126,006 | 52% |
| | | Rehabilitation facilities | 40,580 | 30% | 41,835 | 30% | 43,129 | 29% | 44,463 | 29% | 45,838 | 29% | 47,256 | 29% | 54,136 | 26% | 61,310 | 30% | 67,380 | 32% | 76,888 | 33% | 80,116 | 33% |
| | | Transport costs | 8,028 | 6% | 9,541 | 7% | 11,153 | 8% | 11,738 | 8% | 12,262 | 8% | 13,281 | 8% | 14,971 | 7% | 15,636 | 8% | 16,061 | 8% | 16,692 | 7% | 17,055 | 7% |
| | | Total | 135,423 | 4% | 141,450 | 3% | 146,424 | 3% | 152,137 | 3% | 156,977 | 3% | 162,891 | 3% | 206,019 | 4% | 201,023 | 3% | 211,160 | 3% | 236,402 | 4% | 244,048 | 4% |
| | | Direct cost total | 586,539 | 16% | 641,145 | 15% | 694,537 | 15% | 802,572 | 17% | 859,066 | 17% | 909,002 | 17% | 982,335 | 17% | 1,004,962 | 16% | 1,029,932 | 16% | 1,085,039 | 16% | 1,112,741 | 16% |
| Indirect Cost | | Unemployment | 2,100,191 | 68% | 2,276,720 | 63% | 2,491,115 | 62% | 2,493,093 | 62% | 2,569,126 | 62% | 2,912,681 | 64% | 3,104,954 | 64% | 3,288,018 | 64% | 3,516,994 | 65% | 3,658,302 | 65% | 3,773,508 | 65% |
| | | Reduced productivity at work | 455,272 | 15% | 491,123 | 14% | 537,121 | 13% | 546,017 | 14% | 558,917 | 13% | 524,470 | 11% | 559,134 | 12% | 623,497 | 12% | 622,725 | 12% | 650,899 | 12% | 674,513 | 12% |
| | | Premature mortality | 335,560 | 11% | 510,854 | 14% | 582,374 | 15% | 593,426 | 15% | 585,936 | 14% | 634,954 | 14% | 621,983 | 13% | 659,451 | 13% | 637,711 | 12% | 684,832 | 12% | 682,256 | 12% |
| | | Caregiver productivity loss | 202,672 | 7% | 315,775 | 9% | 396,035 | 10% | 410,914 | 10% | 446,588 | 11% | 505,304 | 11% | 546,587 | 11% | 583,560 | 11% | 614,523 | 11% | 637,921 | 11% | 663,570 | 11% |
| | | Indirect cost total | 3,093,695 | 84% | 3,594,472 | 85% | 4,006,646 | 85% | 4,043,449 | 83% | 4,160,567 | 83% | 4,577,408 | 83% | 4,832,658 | 83% | 5,154,527 | 84% | 5,391,954 | 84% | 5,631,954 | 84% | 5,793,848 | 84% |
| | | Total cost | 3,680,234 | | 4,235,617 | | 4,701,183 | | 4,846,021 | | 5,019,633 | | 5,486,410 | | 5,814,993 | | 6,159,489 | | 6,421,886 | | 6,716,993 | | 6,906,589 | |

in 2016. The direct non-healthcare cost rose from about 135 billion Won in 2006 to 244 billion Won in 2016. The total indirect cost was 3.09 trillion Won in 2006 and increased to 5.79 trillion Won in 2016. Moreover, the direct healthcare cost, direct non-healthcare cost, and indirect cost accounted for about 12–14%, 3–4%, and 83–85% of the total COI of schizophrenia during this period, respectively.

## Total direct and indirect cost of schizophrenia per sex and age group

Fig 1 displays the direct and indirect cost trend of different age groups per sex. The direct healthcare costs of those in their 40s and 50s (both men and women) showed the highest proportion among different age groups. Especially, the direct cost attributable to those in their 50s was the highest among the different age groups since 2014 for men and since 2012 for women. Moreover, individuals who were younger than 20 years and those in their 20s had a much lower proportion of cost compared to those in their 30s, 40s, and 50s.

The indirect cost for all age groups showed a similar pattern for men and women. However, costs attributable to men were higher than among women—especially those in their 40s and 50s. The highest indirect cost was attributable to those in their 40s for both men and women.

## Trend of indirect cost of men and women for each category

Fig 2 shows the trend of indirect cost for men and women per the three cost categories. For both men and women, most of the cost was due to unemployment (about 65% each year). Men showed much higher economic cost due to unemployment than did women. For

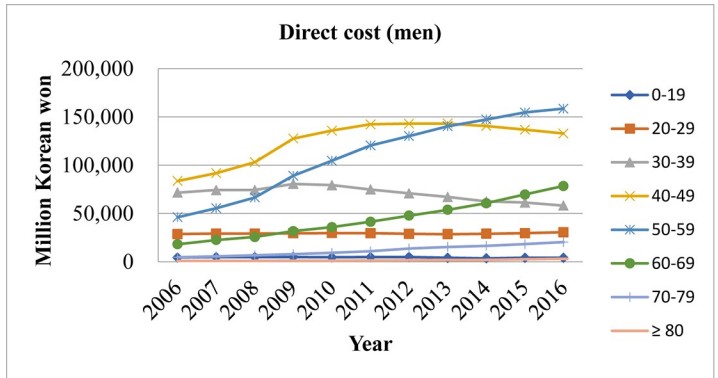
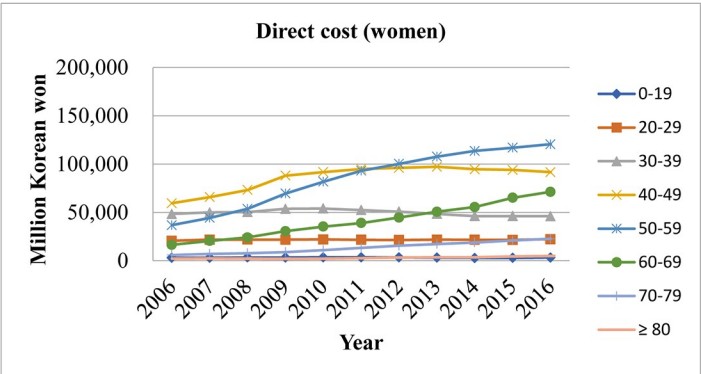
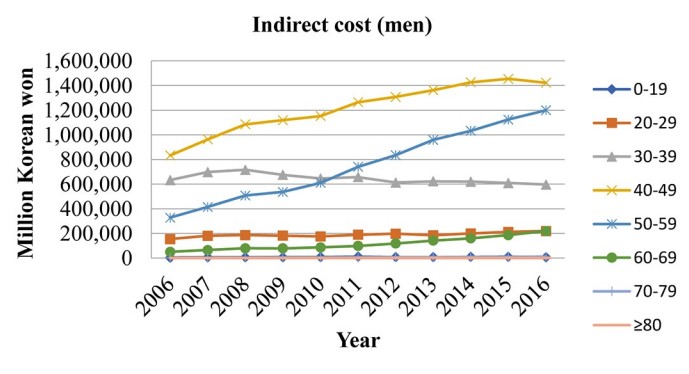
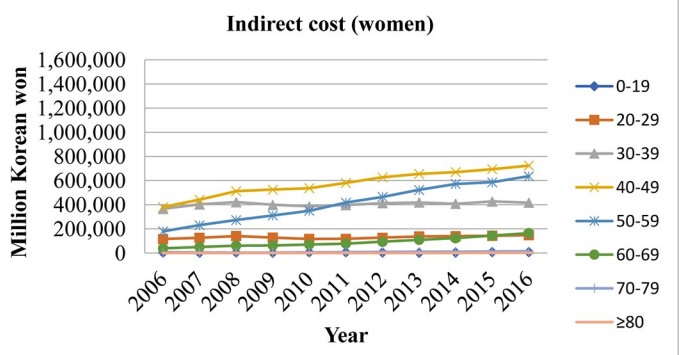

**Fig 1. Direct and indirect cost of men and women by different age groups.**

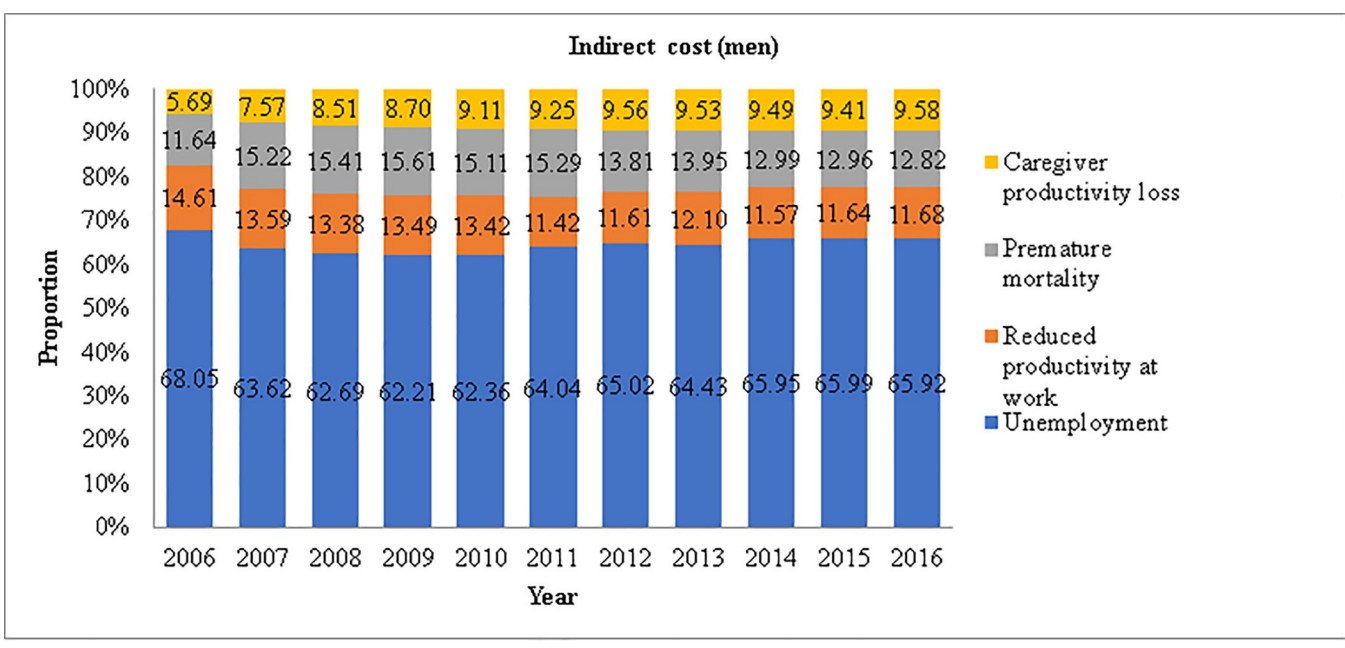

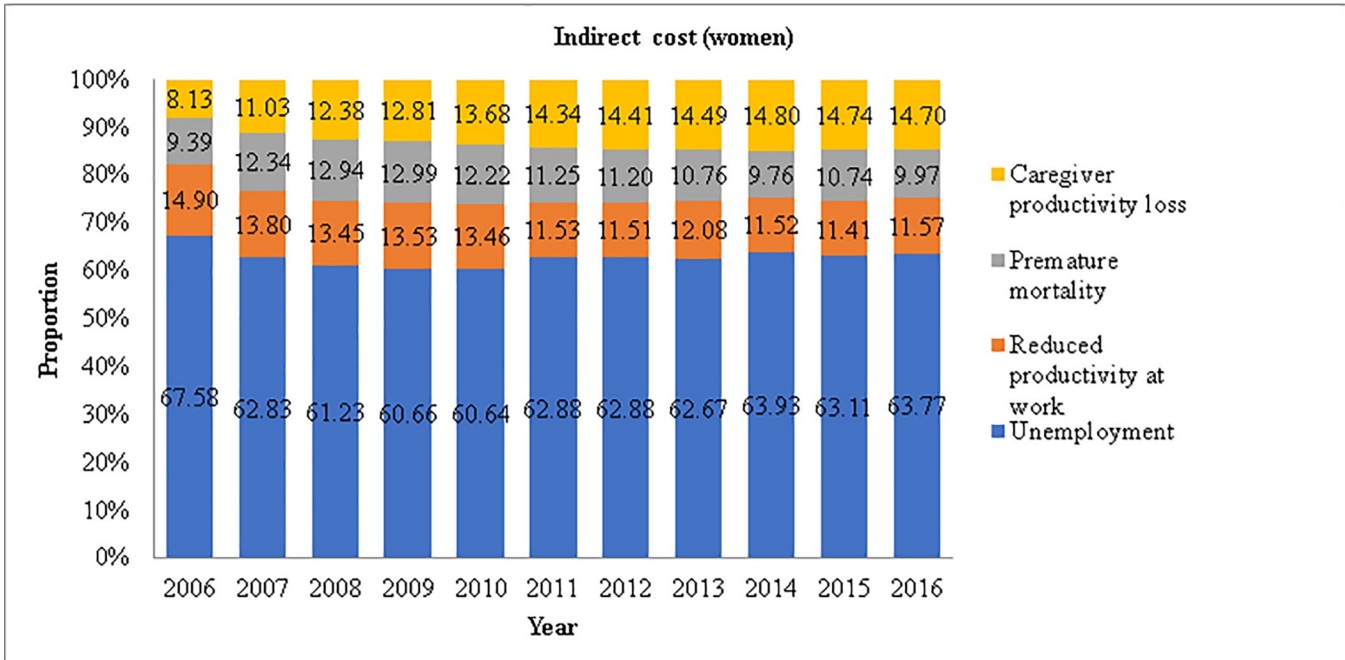

**Fig 2. Indirect cost of men and women by specific costs.**

example, cost due to unemployment for men in 2016 was about 2.42 trillion Won; however, for women, it was 1.35 trillion Won.

## COI trend of schizophrenia in Korea from 2006 to 2016

The COI of treated schizophrenia in South Korea from 2006 to 2016 is presented in Fig 3. The COI of schizophrenia in Korea was 3.68 trillion Won in 2006 and 6.90 trillion Won in 2016, showing a steady increase. Compared to 2006, the treated prevalence and the socioeconomic

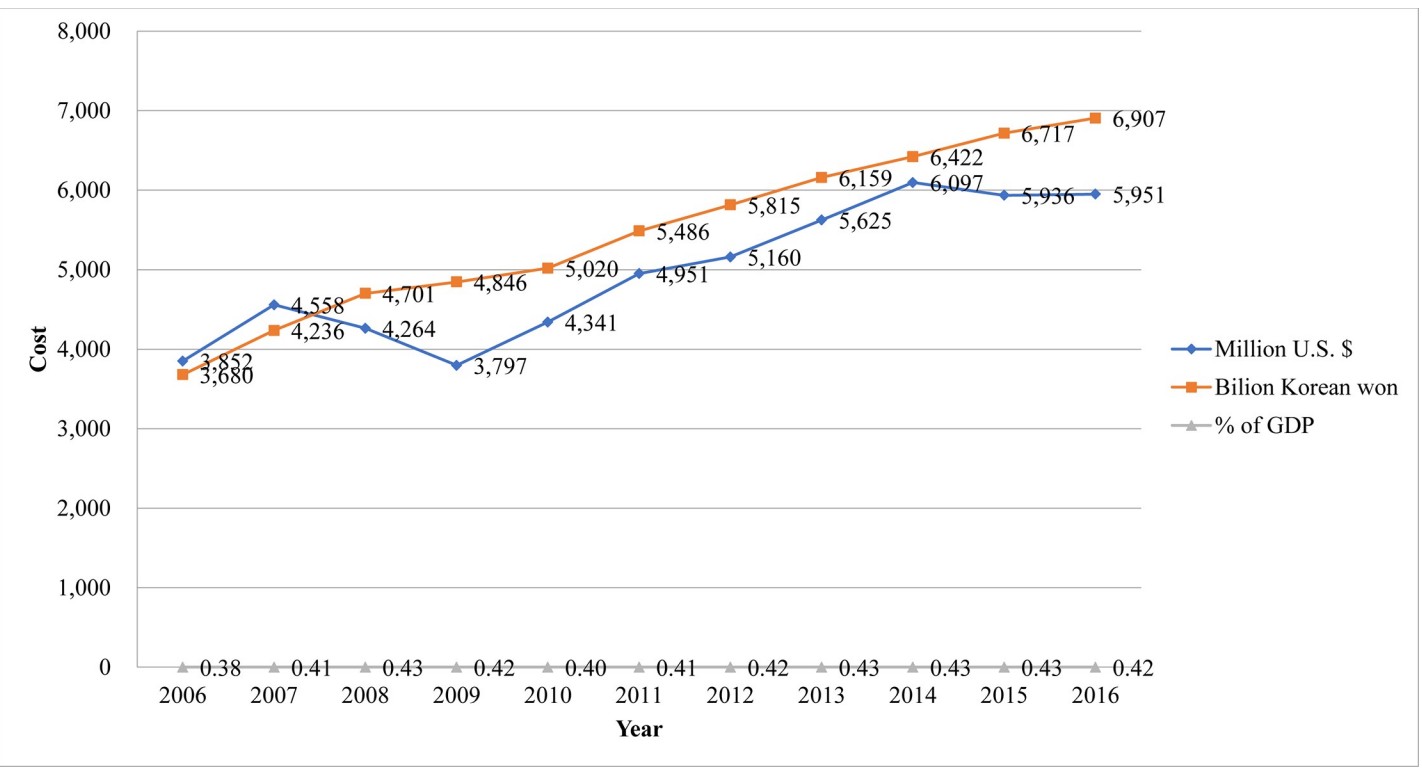

**Fig 3. Economic burden of schizophrenia in Korea from 2006–2016.**

cost of schizophrenia increased by 1.26-fold and 1.88-fold, respectively (whereas Korea's gross domestic product has increased by 1.70-fold) [32]. The COI of schizophrenia in 2016 was 0.42% of Korea's total gross domestic product.

## Sensitivity analysis

The base case COI estimates and the sensitivity analysis results from 2006 to 2016 are shown in Table 3. The COI in 2016 was 6.97 and 7.06 trillion Won when non-covered payments were estimated to be 11% and 20%, respectively. When 0% and 5% discount rates were utilized, the COI of schizophrenia increased to 7.27 trillion Won and decreased to 6.77 trillion Won, respectively.

## Discussion

From a societal perspective, we estimated the COI trend of schizophrenia in Korea from 2006 to 2016. The overall economic burden of schizophrenia has increased steadily from 2006, and the cost trend is similar to that of a previous study that utilized data from 2005 [18].

The increased treatment prevalence could be attributable to Korea's efforts to alleviate stigma against schizophrenia, which led some patients to actively seek help for their symptoms. The term for schizophrenia in Korean was originally "*jungshinbunyeolbyung*," which translates to "mind-split disorder." To reduce stigma, in 2012, the Korean Neuropsychiatric Association changed the term to "*johyeonbyung*," which translates to "attunement disorder." According to a Korean study [33], changing the term had a significant effect on reducing stigma toward schizophrenia.

Table 3. Sensitivity analysis on key burden of illness assumptions.

| Assumption | | | Cost | | | | | | | | | | |
|---|---|---|---|---|---|---|---|---|---|---|---|---|---|
| | | | 2006 | 2007 | 2008 | 2009 | 2010 | 2011 | 2012 | 2013 | 2014 | 2015 | 2016 |
| Base case | | | 3,680,233,795,244 | 4,235,617,109,128 | 4,701,182,922,976 | 4,846,020,781,817 | 5,019,633,100,110 | 5,486,410,086,705 | 5,814,992,976,769 | 6,159,488,759,051 | 6,421,885,598,523 | 6,716,992,852,444 | 6,906,588,614,709 |
| Self-payment cost due to exclusion | 11% | | 3,711,257,139,215 | 4,269,731,656,007 | 4,738,196,072,152 | 4,891,400,747,236 | 5,069,180,707,246 | 5,539,122,026,917 | 5,869,633,553,680 | 6,216,208,548,882 | 6,479,799,073,198 | 6,776,839,043,580 | 6,967,790,505,589 |
| | 20% | | 3,758,443,405,139 | 4,321,639,376,898 | 4,794,535,810,751 | 4,960,780,852,854 | 5,144,995,017,811 | 5,619,920,613,540 | 5,953,597,774,746 | 6,303,349,096,437 | 6,568,803,055,861 | 6,868,910,792,248 | 7,061,921,021,206 |
| Discount rate | 0% | | 3,874,440,653,050 | 4,536,378,779,999 | 5,036,764,090,884 | 5,184,696,012,249 | 5,348,497,726,623 | 5,841,530,537,553 | 6,156,700,946,741 | 6,518,794,425,616 | 6,763,938,237,563 | 7,097,991,969,143 | 7,265,716,631,374 |
| | 5% | | 3,608,316,562,974 | 4,124,713,574,902 | 4,576,781,509,309 | 4,719,882,859,041 | 4,896,913,092,329 | 5,353,699,002,535 | 5,686,672,763,369 | 6,024,025,376,641 | 6,292,603,171,235 | 6,574,780,651,454 | 6,770,256,188,743 |
| Friction period for worker replacement | 1 mo. | | 3,346,061,583,850 | 3,726,839,993,401 | 4,121,236,220,694 | 4,255,075,036,749 | 4,436,206,798,623 | 4,854,178,186,053 | 5,195,726,556,241 | 5,502,927,031,116 | 5,787,023,551,063 | 6,035,133,811,143 | 6,227,442,843,374 |
| | 3 mo. | | 3,348,836,444,680 | 3,730,993,735,545 | 4,126,091,147,815 | 4,260,034,765,749 | 4,441,227,022,623 | 4,859,621,923,053 | 5,201,159,455,241 | 5,508,706,562,116 | 5,792,721,578,063 | 6,041,078,895,143 | 6,233,663,267,374 |
| | 6 mo. | | 3,352,998,735,925 | 3,737,224,348,761 | 4,133,373,538,496 | 4,267,474,359,249 | 4,448,757,358,623 | 4,867,787,528,553 | 5,209,308,803,741 | 5,517,375,858,616 | 5,801,268,618,563 | 6,049,996,521,143 | 6,242,993,903,374 |

mo. = month

Despite the increased treatment prevalence, there is much room for improvement. Among the general population, about 10% received help by a mental health professional (e.g., psychiatrists, other medical doctors, other mental health professionals, nurses, etc.) for mental health problems in Korea [2]. Moreover, only about 20% among those diagnosed with a mental disorder received help from a mental health professional [2], which is much lower than the rate observed in New Zealand (38.9%), the U.S. (43.1%), Spain (35.5%), Belgium (39.5%), and Australia (34.9%) [34–36]. Moreover, a previous study [37] emphasized the need for education on etiology and treatment regarding schizophrenia in Korea to guide patients to use mental health services. Further, sex and personality traits should be considered when providing education to promote help-seeking behavior [38]. Researchers and practitioners thus need to work to raise public awareness concerning schizophrenia to promote patients' help-seeking behavior.

Based on our estimation, inpatient care comprised most of the direct healthcare costs. The inpatient care cost was 75% of the direct healthcare cost in 2006, and this increased to 81% in 2016. The number of admissions into mental hospitals decreased in developed countries [39]; however, Korea shows an opposite trend. The numbers of psychiatric beds have continuously increased and the length of stay at a psychiatric hospital remains long (i.e., 116 days)—in fact, the longest among Organization for Economic Cooperation and Development countries [40]. This could be explained by the payment system in Korea. About 97% of the total population in Korea is supported by the NHIS, and the remaining 3% are supported by Medical Aid. The zero co-payment for those who receive Medical Aid may provoke those to stay longer at a psychiatric hospital. In fact, only 3% of the total population receives Medical Aid; however, about 50% and 60% of the psychiatric admissions and total long-term admissions, respectively, are by those who receive Medical Aid [41]. The government needs to guide these patients to receive treatment from a community-based facility rather than obtaining hospital-based treatment.

Further, the Mental Health Act could account for the high inpatient care costs. Korea has a very high rate (i.e., 80%) of involuntary admissions, which is regulated by the Mental Health Act. This high rate of involuntary admissions is higher than other countries [42]. According to Article XXIV of the Mental Health Act, an institution can involuntarily hospitalize a patient when a psychiatrist judges that he or she needs hospitalization [43]. However, a stricter regulation regarding involuntary admissions and a shorter period before being able to be discharged have been applied owing to the revised Mental Health Act in 2016 [44]. Based on this revision, it is expected that inpatient costs will decrease, and outpatient costs and non-healthcare costs will increase.

Of the different age groups, those in their 40s and 50s accounted for most of the direct costs. Those in their 20s and younger had much lower direct healthcare costs comparatively. The onset of schizophrenia typically occurs in late adolescence [45]. However, our data showed that adolescents and those in their 20s are not actively being treated for schizophrenia. In fact, Korea has a higher duration of untreated psychosis (DUP; i.e., 840 days) compared to Western countries [46]. This indicates that patients are not being treated during the most critical period. Since there is an association between longer DUP and worse outcomes among patients with a psychosis [47], researchers and policymakers need to provide ways to promote patients to seek help when they notice their first psychotic symptom. Especially, an integrated community program is needed to improve outcomes and to raise intervention efficiency [48].

Most direct non-healthcare costs were for sanatoria and rehabilitation facility costs, which accounts for about 90% when combined. This shows that most of the burden from schizophrenia concerns long-term care, which is consistent with previous studies [18,49]. It seems that Korean patients miss the chance for early intervention owing to a high DUP [46], which leads to a poor prognosis [47] and patients requiring inpatient care. Incarceration accounts for about 1% of the direct non-healthcare costs. However, considering the severity of the matter,

efforts to prevent violence that is associated with schizophrenia are needed. A meta-analysis [9] that investigated the association between schizophrenia and violence stated that substance abuse comorbidity mediates the association between schizophrenia and violence. Therefore, experts need to consider preventing substance abuse among patients with schizophrenia.

The cost due to unemployment was the highest among all indirect costs, regardless of sex. However, the cost was much higher for men, which may be attributable to the income difference between men and women in Korea [22]. Moreover, cost due to premature mortality was the second highest for men. This is consistent with previous research showing that men have a higher suicide completion rate than do women [50]. Therefore, suicide prevention interventions are required for men with schizophrenia. For women, the second highest cost among indirect costs was due to caregiver productivity loss since 2010.

Overall, the COI of schizophrenia in Korea has been steadily increasing since 2006. The Korean government has made strides concerning citizens' mental health. The most prominent change is the revised Mental Health Act in 2016 [44]. Through this revision, the unnecessarily long and involuntary admissions of psychiatric patients are expected to decrease. Moreover, the government, researchers, and policymakers need to focus on guiding patients to receive treatment, especially at a community-based facility. Future study is needed to determine how the COI in Korea changes owing to the revision of the Mental Health Act and other efforts the Korean government has implemented.

Despite its importance, this study had some limitations. First, we utilized the treated prevalence rate. Therefore, we did not acquire information about those who have schizophrenia but are not being treated. However, rather than using self-report data, we utilized actual claim data from the NHIS. Second, the costs, especially the indirect costs, are estimated values; the indirect costs could be underestimated or overestimated. For example, the budget for the central government was considered when estimating costs for community centers and specific proportions of budget allocated in each local government to community center were not regarded. Despite these limitations, we tried to use exact numbers when the information was retrievable. For instance, the numbers concerning those with a mental disability, suicide, and mortality were exact numbers rather than estimates.

## Conclusions

The current study estimated the COI trend of schizophrenia in South Korea from 2006 to 2016. This study was based on a societal perspective and we employed a prevalence-based approach for direct costs and a human capital approach for indirect costs. We also utilized information from various sources. The results indicated that more patients with schizophrenia are seeking help; however, there is still room for improvement. Most costs were attributed from indirect costs, and those in their 40s and 50s accounted for most of the COI of schizophrenia in Korea. Our results also imply that community-based care, instead of hospital care, is needed in Korea. Future studies need to examine how the Mental Health Act in 2016 has changed the COI of schizophrenia in Korea.

## Supporting information

**S1 Table. Equations for indirect costs.**
(DOCX)

## Author Contributions

**Conceptualization:** Minkyung Jo, Hyun-Jin Kim, Chul Eung Kim, Subin Park.

**Data curation:** Minkyung Jo.

**Formal analysis:** Minkyung Jo, Hyun-Jin Kim, Min Geu Lee.

**Investigation:** Minkyung Jo.

**Methodology:** Minkyung Jo, Hyun-Jin Kim, Soo Jung Rim, Chul Eung Kim, Subin Park.

**Project administration:** Minkyung Jo, Subin Park.

**Supervision:** Chul Eung Kim, Subin Park.

**Validation:** Hyun-Jin Kim, Subin Park.

**Visualization:** Minkyung Jo, Subin Park.

**Writing – original draft:** Minkyung Jo, Soo Jung Rim, Subin Park.

**Writing – review & editing:** Soo Jung Rim, Subin Park.

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
