## [Editor Report · Decision Letter 0]

2 Oct 2019

PONE-D-19-22006

The cost-of-illness trend of schizophrenia in South Korea from 2006 to 2016

PLOS ONE

Dear Dr Park,

Thank you for submitting your manuscript to PLOS ONE. After careful consideration, we feel that it has merit but does not fully meet PLOS ONE’s publication criteria as it currently stands. Therefore, we invite you to submit a revised version of the manuscript that addresses the points raised during the review process.

Editor comments:

Before submitting the manuscript to peer review, some changes are necessary. In particular:

At the Introduction, please include a brief description of the South Korean health care system. Please, justify the classification of costs. Usually, non-healthcare direct costs include formal and informal care; why are they considered as indirect costs? Also, why community mental healthcare center costs are not considered health care costs?

Please, provide further detail about how the results are obtained (not only about data sources, but about the procedure to arrive to final figures which are shown in the manuscript). For instance, according to reference [10], “Schizophrenia and other psychoses are associated with violence and violent offending, particularly homicide. However, most of the excess risk appears to be mediated by substance abuse comorbidity”. How do you estimate the incarceration costs due to schizophrenia? Do you assume that all the incarceration costs corresponding to schizophrenic inmates are part of non-healthcare direct costs of schizophrenia? This is only an example, more detail should be given for the rest of calculations.

The approach used to estimate indirect costs is the human capital approach, not the prevalence approach. Please, revise the Abstract, the Introduction and the Discussion sections accordingly.

We would appreciate receiving your revised manuscript by Nov 16 2019 11:59PM. To enhance the reproducibility of your results, we recommend that if applicable you deposit your laboratory protocols in protocols.io, where a protocol can be assigned its own identifier (DOI) such that it can be cited independently in the future. For instructions see: http://journals.plos.org/plosone/s/submission-guidelines#loc-laboratory-protocols

We look forward to receiving your revised manuscript.

Kind regards,

Rosa Maria Urbanos Garrido, PhD

Academic Editor

PLOS ONE
---

## [Author Response · Author response to Decision Letter 0]

10 Dec 2019

We provided a brief description of the South Korean healthcare system in the Introduction as follows:

Korea provides universal healthcare through National Health Insurance (NHI) and Medical Aid program. The NHI covers about 97% of the population and the rest is covered by Medical Aid. Consequently, it is possible to construct a health insurance database that contains key data such as health records, prescriptions, etc. Moreover, the NHI Service (NHIS) provides these data to researchers for research purposes. (pg.4) 

We included cost classifications as follows: 

Direct costs are disease-related payments which are divided into two parts: direct healthcare costs and direct non-healthcare costs. Indirect costs account for resources that are lost owing to the disease. (pg. 4)

Direct healthcare costs are defined as medical care costs, which include inpatient care costs, outpatient treatment costs, pharmacy costs, and non-covered care costs. (pg. 5) 

Direct non-healthcare costs are defined as costs that are associated with patient management, other than direct healthcare costs. (pg. 5)

Patients do not receive a diagnosis or prescription at a community mental health center; rather, patients visit community mental healthcare centers for rehabilitation purposes. In addition, most community mental healthcare costs are covered by the government, not by the individual. Therefore, community mental healthcare center costs were considered as direct non-healthcare costs. (pg. 6)

The community mental healthcare center costs, sanatoria costs, and psychiatric rehabilitation costs were estimated by the annual operating costs from each report (e.g., annual operating cost of community mental healthcare center by local mental health centers in 2007 * treated prevalence rate of schizophrenia in 2007). We estimated the transport cost by multiplying the total number of hospital visits by the mean return fare [17]. (pg. 6)

For the exact equations for indirect costs, see Supporting information. (pg. 7 and pg. 26)

The cost-of-illness trend was estimated from a societal perspective using a prevalence-based approach for direct costs and a human capital approach for indirect costs. (pg. 2) 

The prevalence approach was utilized for direct costs, and the human capital approach was used for indirect costs. (pg. 4) 

This study was based on a societal perspective and we employed a prevalence-based approach for direct costs and a human capital approach for indirect costs. (pg. 16)

We revised the Abstract, Introduction, and the Discussion per your request.

---

## [Decision Letter · Decision Letter 1]

11 May 2020

PONE-D-19-22006R1

The cost-of-illness trend of schizophrenia in South Korea from 2006 to 2016

PLOS ONE

Dear Dr Park,

Thank you for submitting your manuscript to PLOS ONE. After careful consideration, we feel that it has merit but does not fully meet PLOS ONE’s publication criteria as it currently stands. Therefore, we invite you to submit a revised version of the manuscript that addresses the points raised during the review process.

In order to provide a more complete information to our readers on the topic, we would like to emphasize the importance to cross referencing very recent material on the same topic published in "PLoS ONE ". Therefore, it would be highly appreciated if you would check the contents published in the last two years of "PLoS ONE" (https://journals.plos.org/plosone/) and add all material relevant to your article to the reference list.

We would appreciate receiving your revised manuscript by Jun 25 2020 11:59PM. To enhance the reproducibility of your results, we recommend that if applicable you deposit your laboratory protocols in protocols.io, where a protocol can be assigned its own identifier (DOI) such that it can be cited independently in the future. For instructions see: http://journals.plos.org/plosone/s/submission-guidelines#loc-laboratory-protocols

We look forward to receiving your revised manuscript.

Kind regards,

Wen-Jun Tu

Academic Editor

PLOS ONE

Journal Requirements:

1. Please ensure that all statement are supported by adequate references, and that the references studies are critically analysed. For example, we note that your statement "Patients with schizophrenia are involved in crime more so than is the general population" does not seem to reflect the conclusions drawn by the study referenced ( Fazel, Seena, et al. "Schizophrenia and violence: systematic review and meta-analysis." *PLoS medicine* 6.8) , which concluded that "most of the excess risk appears to be mediated by substance abuse comorbidity". Please ensure that the aspect of substance abuse comorbidity is mentioned, to provide a balanced view on the association between schizophrenia and crime prevalence, and to reflect the content of the cited study. 

Additional Editor Comments (if provided):

Reviewers' comments:

Reviewer's Responses to Questions

**Comments to the Author**

1. If the authors have adequately addressed your comments raised in a previous round of review and you feel that this manuscript is now acceptable for publication, you may indicate that here to bypass the “Comments to the Author” section, enter your conflict of interest statement in the “Confidential to Editor” section, and submit your "Accept" recommendation.

Reviewer #1: All comments have been addressed

2. Is the manuscript technically sound, and do the data support the conclusions?

Reviewer #1: Yes

3. Has the statistical analysis been performed appropriately and rigorously? 

Reviewer #1: Yes

4. Have the authors made all data underlying the findings in their manuscript fully available?

Reviewer #1: Yes

5. Is the manuscript presented in an intelligible fashion and written in standard English?

Reviewer #1: Yes

6. Review Comments to the Author

Reviewer #1: (No Response)

7. PLOS authors have the option to publish the peer review history of their article (what does this mean?). If published, this will include your full peer review and any attached files.

Reviewer #1: Yes: Silvia Coretti

---

## [Author Response · Author response to Decision Letter 1]

17 Jun 2020

We greatly appreciate the academic editor and reviewer for thoroughly reviewing our manuscript. We made revisions per the comments and suggestions that were provided, which are highlighted below. We hope our responses are suitable and that the revised manuscript is now acceptable for publication in PLoS One.

Journal Requirements 

1. Please ensure that all statement are supported by adequate references, and that the references studies are critically analysed. For example, we note that your statement "Patients with schizophrenia are involved in crime more so than is the general population" does not seem to reflect the conclusions drawn by the study referenced ( Fazel, Seena, et al. "Schizophrenia and violence: systematic review and meta-analysis." PLoS medicine 6.8) , which concluded that "most of the excess risk appears to be mediated by substance abuse comorbidity". Please ensure that the aspect of substance abuse comorbidity is mentioned, to provide a balanced view on the association between schizophrenia and crime prevalence, and to reflect the content of the cited study.

Thank you for providing this important comment. We clarified the association between schizophrenia and violence in the revised manuscript as follows:

- Patients with schizophrenia are involved in crime (mediated by substance use comorbidity) more so than is the general population [9]. (pg. 5, lines 114–116)

- Incarceration accounts for about 1% of the direct non-healthcare costs. However, considering the severity of the matter, efforts to prevent violence that is associated with schizophrenia are needed. A meta-analysis [9] that investigated the association between schizophrenia and violence stated that substance abuse comorbidity mediates the association between schizophrenia and violence. Therefore, experts need to consider preventing substance abuse among patients with schizophrenia. (pg. 15, lines 278–284)

2. In order to provide a more complete information to our readers on the topic, we would like to emphasize the importance to cross referencing very recent material on the same topic published in "PLoS ONE ". Therefore, it would be highly appreciated if you would check the contents published in the last two years of "PLoS ONE" (https://journals.plos.org/plosone/) and add all material relevant to your article to the reference list.

We went through recent materials related to our topic in PLoS One and added one relevant study to our reference list, which revealed that patients with schizophrenia have a high risk of acquiring physical diseases (e.g., cardiovascular disease):

- [10] Cunningham R, Poppe K, Peterson D, Every-Palmer S, Soosay I, Jackson R. Prediction of cardiovascular disease risk among people with severe mental illness: a cohort study. PLoS One. 2019;14(9): e0221521.

---

## [Editor Report · Decision Letter 2]

23 Jun 2020

The cost-of-illness trend of schizophrenia in South Korea from 2006 to 2016

PONE-D-19-22006R2

Dear Dr. Park,

We’re pleased to inform you that your manuscript has been judged scientifically suitable for publication and will be formally accepted for publication once it meets all outstanding technical requirements.

Kind regards,

Wen-Jun Tu

Academic Editor

PLOS ONE
---

## [Editor Report · Acceptance letter]

1 Jul 2020

PONE-D-19-22006R2 

The cost-of-illness trend of schizophrenia in South Korea from 2006 to 2016 

Dear Dr. Park:

I'm pleased to inform you that your manuscript has been deemed suitable for publication in PLOS ONE. Congratulations! Your manuscript is now with our production department. 

Kind regards, 

on behalf of

Dr. Wen-Jun Tu 

Academic Editor

PLOS ONE